# Complex Study of Eutectoidal Phase Transformation of 2507-Type Super-Duplex Stainless Steel

**DOI:** 10.3390/ma12132205

**Published:** 2019-07-09

**Authors:** István Mészáros, Bálint Bögre

**Affiliations:** Department of Materials Science and Engineering, Budapest University of Technology and Economics, Bertalan Lajos utca 7, 1111 Budapest, Hungary

**Keywords:** duplex stainless steel, eutectoidal decomposition, cold rolling, heat treatment, magnetic testing, thermoelectric power, EBSD, activation energy

## Abstract

The aim of this work was to study expansively the process of the eutectoidal phase transformation of 2507-type super-duplex stainless steel. Three sample sets were prepared. The first sample set was made to investigate the effect of the previous cold rolling and heat treatment for the eutectoidal phase transformation. Samples were cold rolled at seven different rolling reductions which was followed by heat treatment at five different temperatures. The second sample set was prepared to determine the activation energy of the eutectoidal decomposition process using the Arrhenius equation. Samples were cold rolled at seven different rolling reductions and were heat treated at the same temperature during eight different terms. A third sample set was made to study how another plastic-forming technology, beside the cold rolling, can influence the eutectoidal decomposition. Samples were elongated by single axis tensile stress and were heat treated at the same temperature. The results of the first and the third sample sets were compared. The rest δ-ferrite contents were calculated using the results of AC and DC magnetometer measurements. DC magnetometer was used as a feritscope device in this work. Light microscope and electron back scattering diffraction (EBSD) images demonstrated the process of the eutectoidal decomposition. The thermoelectric power and the hardness of the samples were measured. The results of the thermoelectric power measurement were compared with the results of the δ-ferrite content measurement. The accurate value of the coercive field was determined by a Foerster-type DC coercimeter device.

## 1. Introduction

Duplex stainless steel (DSS) has a double-phase microstructure containing, in approximate equal proportion, ferrite and austenite. The double-phase structure causes an excellent combination of strength and corrosion resistance, mainly in chloric medium. DSS is used mainly in chemical processing and transport, oil and gas refining, paper manufacturing, and in the marine environment [1,2,3].

Table 1 shows the characteristic chemical composition of DSS, which provides the favorable properties of this type stainless steel [3].

The carbon content of DSS is strictly limited because of corrosion resistance. The high chromium alloying increases the corrosion resistance until the nickel improves the toughness. The nitrogen increases the strength and resistance against the pitting corrosion.

In the double-phase structure, the austenite-forming elements (C, Ni, N, Cu) and ferrite-forming elements (Cr, Mo, W) have a well-adjusted ratio, which is the base of the microstructural equilibrium. The austenite ensures the good ductility, toughness, and weldability until the ferrite raises the corrosion resistance mainly against the pitting, stress, and crevice corrosion. 

Secondary phases can appear during the heat treatment in the critical temperature range, about 300–1000 °C, due to the metastable structure. Higher alloying content (mainly the chromium and molybdenum) can increase the chance of the forming of precipitations. The appearance of these phases are dangerous because these can cause the dramatic decrease of the ductility and the corrosion resistance. Figure 1 shows these typical precipitations of DSS. Among them, the tetragonal σ-phase is the most significant, which can appear in the case of molybdenum alloying in the temperature range 600–1000 °C. χ-phase (Fe_36_Cr_12_Mo_10_) can form between 700–850 °C. The Cr_2_N precipitation has a hexagonal crystal lattice and settles down at the border of the ferrite grains. The high chromium content of Cr_2_N causes chromium impoverishment in its environment which can cause intensified corrosion sensibility. The R-phase forms in the temperature range 550–650 °C and it increases the brittleness. M_23_C_6_ and M_7_C_3_ precipitations are complex carbides with high chromium content. The above characterized precipitations appear at high temperature and in a relatively short time, as it can be seen in Figure 1 [1,2,3,4,5,6,7].

The temperature range of the brittle π-phase (Fe_7_Mo_13_N_4_) is at 550–600 °C. ε-phase (Fe_3_N and Fe_2_N) which is a nitride similarly to the π-phase can be noticed in copper-alloyed DSS. α’-phase has the most notable effect at low temperature. This precipitation causes the 475 °C embrittlement and ferritic zones which are rich in chromium. 

The most significant phase transformation in duplex stainless steel is the eutectoidal decomposition of δ-ferrite while δ-ferrite transforms into σ-phase and secondary austenite (δ → σ + γ_2_) [1,2,3,4,5,6,7].

Figure 2 represents G. Herbstleb and P. Schwaab’s simplified precipitation diagram for DSS [3]. M_23_C_6_-type complex carbides with high chromium content form at the border of the δ-ferrite grains (δ/δ) at the early stage of the decomposition of δ-ferrite. The appearance of the M_23_C_6_-type carbides passes any other phase transformation due to the high mobility of the carbon, as Figure 2 illustrates. The carbon atoms diffuse to the grain boundary because the enrichment of the carbide-forming alloys (mainly chromium and molybdenum) are significant at the boundary. These carbides can be favorable places for the later-forming σ-phase and secondary austenite.

Figure 3 illustrates the kinetics of the eutectoidal decomposition of δ-ferrite [1,2,3,4,5,6,7]. The distribution of the alloying elements is not homogeneous in DSS due to its dual phase structure. The δ-ferrite is richer in ferrite- and carbide-forming elements (chromium and the molybdenum) and poorer in austenite-forming elements (nickel and nitrogen). The distribution of the above-mentioned alloying elements is opposite in austenite. Due to heat treatment the chromium and molybdenum diffuse to the grain boundary of the δ-ferrite/austenite and form σ-phase precipitations. Meanwhile the δ-ferrite becomes poor in these alloying elements, loses its stability, and transforms into secondary austenite [8].

The above-presented description showed how complex metallurgical processes can take place in DSS. The application temperature of DSS is limited at maximum 280–325 °C due to the appearance of the undesirable precipitations [3,9,10,11,12,13,14,15,16,17,18].

DSS has four grades with different pitting resistance equivalent numbers (PREN) which can characterize the corrosion resistance of the steel: lean DSS (PREN < 35), standard DSS (PREN 35–40), super-duplex stainless steel (SDSS) (PREN 40–45), and hyper-DSS (PREN > 45) [3,9,10,11,12,13,14,15,16,17,18].

The aim of this paper is to study, complexly, how the previous cold rolling can influence the eutectoidal decomposition of δ-ferrite in 2507 SDSS during the heat treatment. 

## 2. Material and Sample Preparation 

Table 2 and Table 3 show the nominal chemical composition and the main mechanical properties of the 2507-type SDSS. The main alloying elements are the chromium (about 25%) and the nickel (about 7%).

Figure 4 represents the original sheet material and the directions of the manufacturing hot rolling and the experimental cold rolling. Samples were cut from sheet material with a band saw. The thickness (h_0_) was about 10 mm, the width (w_0_) was about 15 mm, and the length (l_0_) was about 100 mm of the cut samples.

The samples were cold rolled by a Ø300 mm diameter double-cylinder rolling machine. The direction of the cold rolling was perpendicular to the direction of the manufacturing hot rolling. The thickness reductions were 0.25 mm in every rolling step.

## 3. Test Results and Discussion

### 3.1. First Sample Set

The first sample set was prepared to study the process of the eutectoidal phase transformation due to the previous cold rolling and heat treatment. The seven rolling reductions (ε) were the following: 0%, 10.3%, 22.3%, 31.3%, 41.6%, 50.6%, and 61.9%. The rolling reductions were calculated by the Equation (1):(1)$$\varepsilon =\left(h_{0}-h\right)/h_{0}*100\left(\%\right)$$
where “h” was the thickness of the rolled sample. Five samples were rolled from every rolling reduction and were heat treated at 20, 700, 750, 800, and 850 °C temperatures. The term of the heat treatment was 30 min and the samples were normalized on static normal air. Naturally, the different rolling reductions resulted in different sizes of the samples. At the end of the preparation process all samples were machined at the same size 3.4 mm × 10 mm × 100 mm (h × w × l).

#### 3.1.1. AC Magnetometer Measurement 

In the investigated DSS, the δ-ferrite is the only ferromagnetic phase which transforms to paramagnetic σ-phase and secondary austenite. Therefore, the eutectoidal decomposition influences the ferromagnetic phase ratio of the alloy. It is well known that the magnetic saturation polarization is linearly proportional to the ferromagnetic phase ratio of alloys [19]. Consequently, the δ-ferrite ratio can be precisely determined from saturation polarization. 

Firstly, the samples were measured by an AC magnetometer to determine their δ-ferrite content. Figure 5 illustrates the set-up of the AC magnetometer [20] which was designed and built in our laboratory. This set up is suitable only for measuring flat-stripe-shaped samples.

The instrument measures the normal magnetization curve and the hysteresis loop of the samples. The maximal polarization, coercive field, remnant induction, and initial permeability can be determined from the magnetization curves.

The yoke stands from two symmetrical U-shaped laminated Fe-Si iron cores which closes the magnetic circle. The driving and the pick-up coils close round the sample. The power amplifier and the function generator supply sinusoidal excitation current which frequency is 5 Hz. The 16-bit input–output data acquisition card accomplished the measurements. In case of each sample, 200 minor hysteresis loops were recorded. The maximum excitation field strength was about 128 A/cm which cannot saturate the samples magnetically. Because of this physical limitation, the AC magnetometer is not able to determine the value of saturation polarization. Therefore, the δ-ferrite content was calculated from the measured maximal value of polarization. The δ-ferrite content of the undeformed and non-heat-treated (initial) sample was 46.9% according to the manufacturer data sheet. Its measured maximal polarization ($\mu_{0}M_{max})$ was 0.31 T. The δ-ferrite content of the tested samples (*x* in %) was determined using a simple proportion with the following Equation (2):(2)$$x=\frac{46.9}{0.31}{\left(\mu_{0}M_{max}\right)}_{measured\;sample}$$

The process of the eutectoidal decomposition can be noticed particularly well in Figure 6, which represents the calculated δ-ferrite contents in the function of the heat treatment temperature [21]. 

As it can be seen, the δ-ferrite content of the deformed and non-heat-treated samples is about equal, the cold rolling itself cannot influence the δ-ferrite content. On the other hand, it can be noticed the δ-ferrite content decreases due to the heat treatment by each deformation rate because of the intensifying δ-ferrite transformation. The eutectoidal decomposition starts at about 750 °C and is more intensive in deformed samples. The stronger the extent of the previous cold rolling reduction, the more the amount of the decomposed δ-ferrite. In other words, the previous cold rolling promotes the δ-ferrite decomposition. It is supposed that the deformation stored energy improves the number of the σ-phase nuclei during the heat treatment. Due to the growing amount of the σ-phase, more δ-ferrite grains transform into secondary austenite. 

#### 3.1.2. DC Magnetometer Measurement

Secondly, the δ-ferrite content of the samples was measured by a Stablein-Steinitz type DC magnetometer (designed and made in our department) and were compared with the results of the AC magnetometer. Figure 7 illustrates the set-up of the applied DC magnetometer [22,23]. The advantage of the DC magnetometer against the AC magnetometer is it can excite the samples into saturation. The highest excitation level was about 2700 A/cm, which was enough to reach the complete saturation of DSS samples. The DC magnetometer can measure the real saturation polarization until the AC magnetometer can determine just a maximal polarization value. Unfortunately, the DC magnetometer has disadvantages as well. It requires bulk samples and it can only be used in a laboratory because of its heavy size. The DC magnetometer was designed and built in our laboratory [22,23,24].

The original version of Stablein-Steinitz type DC magnetometer was designed to record the hysteresis curve of bulk materials in the 1930s. It has symmetrical yoke which contains two U-shaped parts and a small cross-section middle bridge. The excitation is accomplished by four coils. The set-up consists of two uniform-sized air-gaps namely, the reference, and the measuring air-gaps. The arrangement is magnetically symmetrical; therefore, there is no flux in the middle bridge if the sample air-gap is empty. The symmetry of the magnetic circuit is broken by a sample taken into the sample air-gap. Therefore, some part of the flux closes through the middle bride. Our set up contained two Hall sensors and a PC-driven data acquisition unit. The sensor in the sample air-gap measures the magnetic field (H) within the sample, the signal of the sensor in the middle bridge is directly proportional to the magnetization (M) of the sample.

The signals of the Hall sensors are connected to the 16-bit data acquisition card through a double-channel amplifier. The excitation is supplied by a computer-controlled power amplifier. The LabView program allows for cyclical demagnetization and it can record, among others, the normal magnetization curve and the hysteresis loops of the sample [22,23,24].

The δ-ferrite content was calculated from the saturation polarization values similarly to the method mentioned before. The measured saturation polarization ($\mu_{0}M_{saturation})$ of the initial sample was 0.563 T. The δ-ferrite content of the tested samples (*x* in %) was determined with the following Equation (3):(3)$$x=\frac{46.9}{0.563}{\left(\mu_{0}M_{saturation}\right)}_{measured\;sample}$$

Figure 8 represents the δ-ferrite content values in function of the heat treatment temperature by the DC magnetometer measurement.

The δ-ferrite content reduces continuously in function of the heat treatment temperature by every rolling reduction. It can be noticed that the δ-ferrite content reduction below 750 °C is less intensive than it is by the AC magnetometer measurement. The DC magnetometer determines higher δ-ferrite contents than the AC magnetometer due to their different excitation levels. Naturally, the calculation which derives the δ-ferrite contents from the real saturation polarization values give more accurate results.

In the following part of this work the DC magnetometer was used as a feritscope to determine the δ-ferrite content of samples.

#### 3.1.3. Light Microscope

All samples were examined by an Olympus PMG-3 type metallographic microscope (Olympus, Hamburg, Germany) which has a digital camera and the maximal magnification is 1000×. During the preparation, the samples were fixed in resin and were grinded on different grain size Al_2_O_3_ grinding papers. After the grinding, the samples were polished on a fine cloth using Al_2_O_3_ suspension. Buehler EcoMet 30-type manual metallographic machine (Buehler, Lake Bluff, IL, USA) was used for the grinding and polishing. The type of etching liquid was Beraha [25].

Images were taken about all samples in the magnification of 25×, 50×, 100×, 500×, and 1000×. Figure 9 shows the microscope images of 2507-type SDSS in the magnification of 1000×. 

Figure 9a shows the original microstructure of the 2507-type SDSS without deformation and heat treatment. The ratio of the δ-ferrite and the austenite is almost equal. The microstructure of the heat treated sample at 850 °C without cold rolling is shown in Figure 9b. It can be noticed that the transformation of the δ-ferrite has already begun, a slight amount of σ-phase appeared at the grain boundary of the δ-ferrite and the austenite. Figure 9c represents the ε = 61.9% deformed samples without heat treatment. It can be seen that the decomposition of the δ-ferrite cannot begin without heat treatment. Sample which was deformed in ε = 61.9% and heat treated at 850 °C is shown in Figure 9d. When the eutectoidal decomposition is finished, the δ-ferrite transformed completely into σ-phase and secondary austenite.

#### 3.1.4. EBSD

The samples were examined by a Philips XL30 ESEM FEG-type scanning electron microscope (SEM, Amsterdam, The Netherlands). The SEM has a point-source cathode of tungsten, which has a surface layer of zirconia (ZrO_2_). The high tension is continuously variable from 0.2 till 30 kV and the size of the specimen stage is 50 mm × 50 mm × 50 mm.

The type and the distribution of the phases were detected by electron back scattering diffraction (EBSD). Figure 10 shows the phase maps of the 2507-type SDSS where the different phases were signed by color codes. The green color means the austenite, the red area shows the δ-ferrite, and the yellow color represents the tetragonal σ-phase [26,27].

The EBSD images can similarly illustrate the eutectoidal decomposition of δ-ferrite to what was presented by the optical microscope examination. Figure 10a shows the original phase ratio: 41% δ-ferrite and 59% austenite. This result is slightly different form the value which is given on the data sheet of the 2507-type SDSS, it represents 46.9% δ-ferrite. Figure 10b shows the phase map of the heat-treated sample at 850 °C without deformation. The ratio of the δ-ferrite decreased from 41% to 35.3% and 3% σ-phase appeared. The amount of the austenite increased from 59% to 61.6%. The decomposition of the δ-ferrite becomes more intense in Figure 10c, in which the sample is deformed in ε = 22.3% and heat treated at 850 °C; 13% σ-phase can be detected beside the δ-ferrite and the austenite. The sample in Figure 10d was prepared with the maximal deformation extent and heat treatment temperature. The phase transformation of the δ-ferrite has almost finished: The amount of the σ-phase increased significantly until the δ-ferrite content is just about 1%.

#### 3.1.5. Thermoelectric Power Measurement

Thermoelectric power (TEP) was measured by a TechLab Trivolt PK120-type TEP measuring instrument (TechLab, La Tannerie, France) in order to examine if there is correlation between the eutectoidal decomposition of δ-ferrite and the TEP. The TEP instrument is operated by 220 V and its measuring accuracy is about 2 nV/K.

The basis of the TEP measurment is the Seebeck-effect. If temperature difference forms between two diverse electrical conductors or semi-conductors, voltage difference appears between the two substances. The value of the TEP is sensitive to the different material properties, especially to the chemical composition [28,29,30]. Figure 11 shows the sematic illustration of the TEP instrument [29]. 

The set-up contains two copper blocks, one of them is heated electrically, while the other is cooled by circulated water. The measuring temperatures can be controlled quickly and exactly due to the thermocouples which are built in the cold and the hot blocks. The thickness of the samples was 3.4 mm. The contact surface of the samples were grinded with fine grinding paper (P1200) and then were cleaned with alcohol. Samples were put on the cold and hot blocks and were fixed with two isolated screws. Temperature of the cold block was about 15 °C until the hot block was about 25 °C. The result was read after about 90 s, when the value of the TEP stabilized. The results of the TEP measurments are presented in Figure 12 and Figure 13.

Figure 12 shows the values of the TEP in function of the heat treatment temperature. It can be seen the values of the TEP are nearly independent of the deformation rate below 750 °C. However, the previous cold rolling has an intensive effect on the TEP above 750 °C. The stronger the plastic deformation rate, the lower the values of the TEP. The changing of the TEP above 750 °C is quite similar to the changing of the δ-ferrite content, which were represented in Figure 6 and Figure 8.

Figure 13 shows the values of the TEP in function of the rolling reduction. It is known that the TEP is sensitive to the precipitations. The progressive decrease of the TEP at 800 and 850 °C can occur due to the precipitation of the significant amount of σ-phase, which forms during the eutectoidal decomposition. It can be seen that the values of the TEP are nearly independent of the deformation rate at 700 and 750 °C, but higher than the values of the non-heat-treated samples. It is considered that the increase of the TEP is caused by those precipitations which appear previously to the σ-phase (e.g., Cr_2_N, M_23_C_6_, or χ-phase) [3]. Based on the above, there is a good correlation between the process of the eutectoidal decomposition and the results of the TEP measurement.

#### 3.1.6. Foerster-Type DC Coercimeter Measurement

The accurate value of the coercive field was measured by a Foerster DC coercimeter (Institut Dr. Förster, Reutlingen 1.094 and 1.106, Reutlingen, Germany) which is an open magnetization circuit equipment. Figure 14 shows the set-up of the DC coercimeter. The equipment contains a solenoid coil and two high-sensitivity magnetic field sensors, which are exactly in the middle outside of the coil.

As a first step (Figure 14a), the sample is put inside in the middle of the solenoid coil and magnetized into saturation. The polarization of the sample is measured by two sensors, which can measure the component of the magnetic field vector which is perpendicular to the coil. The sample creates a magnetic field outside the coil which is proportional with the magnetization of the sample. This magnetic field vector has just a horizontal component at the position of the sensors. As a second step (Figure 14b), the sample is moved horizontally in the solenoid coil until the sensors detect the maximal perpendicular component of the magnetic field vector. A reverse magnetic field is built up with the coil and it is increased until the measured perpendicular field component becomes zero. The reverse magnetic field is equal with the coercive field of the sample. The maximum of the magnetization field was 1000 A/cm.

Figure 15 represents the values of the coercive field in function of the heat treatment temperature. It can be noticed that the coercive field increases progressively in function of the heat treatment temperature by every rolling reduction. The increase is caused by two reasons: the plastic deformation and the appearance of the σ-phase. The coercive field of the non-heat-treated samples rises due to the cold rolling. σ-phase precipitations cause the increment of the coercive field by the undeformed samples. The increase of the coercive field is much higher by the strongly cold rolled samples than it is by the lower-extent deformed samples. The coercive field measurement showed the effect of the plastic deformation and the appearance of the σ-phase is not simply added by the cold rolled and heat treated samples, but the deformation stored energy increases the number of the σ-phase nuclei. It is considered that the reason for the coercive field increment is that the σ-phase precipitations prohibit the movement of the domain walls. The highly deformed samples (ε = 41.6%, 50.6%, and 61.9%) after heat treatment at 850 °C became nearly paramagnetic. Their coercive field cannot be determined.

#### 3.1.7. Hardness Measurement

The Vickers hardness (HV 10) of the samples were measured by a KB 250 BVRZ-type universal hardness testing machine, which was produced by KB Prüftechnik GmbH (Hochdorf-Assenheim, Germany)**.** The measuring limit of the machine is 250 kg and the test room height is 320 mm.

Figure 16 shows the hardness of the samples in function of the heat treatment temperature. The load was nominally 98.07 N during 12 s.

The hardness increase of the undeformed samples is 54 HV until this rise of the strongly rolled samples is three times higher (ε = 50.6% and 61.9%). The increase of the hardness is caused by the dislocation hardening and the σ-phase precipitation, as it was specified in the before chapter. The previous cold rolling before the heat treatment increases the chance of nucleation of the σ-phase along the slip lines. More σ-phases can cause a higher increase in hardness.

### 3.2. Second Sample Set

The second sample set was prepared to determine the activation energy of the eutectoidal decomposition process. The samples were cold rolled and heat treated at 850 °C during different terms. The extents of the rolling reduction were similar to the first sample set: ε = 0%, 10.2%, 21.9%, 29.9%, 40.9%, 50.1%, and 61.1%. The rolled samples were cut into eight smaller pieces and each piece was heat treated separately until the following terms: t = 0, 20, 25, 30, 35, 40, 45, and 50 min. The heat treated samples were normalized using static normal air.

Activation energy was calculated using the Avrami and Arrhenius equations. The activation energy can be considered as the minimal energy which is necessary for the beginning of a reaction or a phase transformation. This energy can describe a phase transformation numerically.

Kinetics equations are used to determine the time of a phase transformation in alloys, of which the most current is the Avrami equation. This equation gives a relationship between the transformed fraction and the time. Equation (4) shows the general form of the Avrami equation:(4)$$y=1-e^{-rt^{n}}$$
where “*y*” is the transformed fraction, “*t*” is the time, and “*n*” and “*r*” are the variables. The exponential connection between the rate of the chemical reaction and the temperature is given by the Arrhenius equation which is showed with Equation (5):(5)$$k=Ae^{-\frac{E_{a}}{RT}}$$
where “*k*” is the rate of transformation, “*A*” is the pre-exponential factor, “*E_a_*” is the activation energy, “*R*” is the universal gas constant, and “*T*” is the absolute temperature. The Arrhenius equation is applied for the description of the temperature dependence of thermally-activated processes [31,32,33].

Figure 17 shows the Avrami curves which were determined from the decomposed δ-ferrite content. The curves were fitted by two parameters regression using the Avrami Equation (4) and the OriginPro 8 software (OriginLab, Northampton, MA, USA).

The rate of the transformation can be determined with the Avrami curves using Equation (6):(6)$$k=\frac{1}{t_{0.5}},$$
where $t_{0.5}$ is the time which belongs to the *y* = 0.5 transformed fraction. Taking the logarithm of Equation (5), the activation energy can be calculated as Equation (7):(7)E=RT(lnA−lnk)

Value of the lnA was determined by the so-called Arrhenius plot using the data of the 61.9% deformed samples of the first sample set. It was supposed that its value is independent of deformation extent. The rate of transformation (*k*) can be calculated from the amount of decomposed δ-ferrite as Equation (8) shows: (8)$$k=\left(F_{0}\%-F\%\right),$$
where “*F*_0_%” is the original δ-ferrite content (46.9%) and “F%” is the calculated δ-ferrite content after the decomposition. Equation (9) shows the replacement of the “*k*” in the natural logarithm of the Arrhenius equation:(9)$$ln\left(F_{0}\%-F\%\right)=lnA-\frac{E_{a}}{R}\frac{1}{T}$$
If “$ln\left(F_{0}\%-F\%\right)$” is plotted in function of the “1/*T*,” the intercept of the line is equal with the value of the lnA. Figure 18 shows the Arrhenius plot of the ε = 61.9% deformed first sample set. The obtained value of lnA was 28.82, which was substituted into Equation (7) for calculating the activation energy (*E*) values of the second sample set.

Figure 19 represents the obtained activation energy values of the second sample set in function of the rolling reduction.

It can be seen that the activation energy decreases from 302 to 296 kJ/mol in function of the rolling reduction. 

It is supposed that the rate limiting process of the δ-ferrite decomposition is the diffusion of Cr and Mo in δ-ferrite according to the kinetics described before. Slightly different activation energy values of Cr and Mo diffusion in ferrite are published in scientific papers. The typical values are 267.4 kJ/mol (Cr in ferrite) and 282.6 kJ/mol (Mo in ferrite) [34,35,36], which are in good agreement with the obtained data.

It should be noticed that the determination of the activation energy with the Arrhenius fitting and Avrami equation is a very sensitive calculation method. If the δ-ferrite content changed 3% by the Arrhenius fitting the calculated activation energy increased almost 30%. The more accurate determination of the activation energy would require numerous samples.

### 3.3. Third Sample Set

The third sample set was prepared to study how the eutectoidal phase transformation is influenced by plastic deformation technology that is different from cold rolling. Samples were elongated by single-axis tensile stress and were heat treated at 850 °C. The samples were in the furnace for 30 min and were normalized using static normal air. The effect of the cold rolling and the elongation for the eutectoidal phase transformation were compared.

The elongation was made by a Heckert EU-40-type hydraulic tensile test machine (Mönchengladbac, Germany), which measuring limit is 400 kN. The machine has a digital data acquisition card and its stroke length is about 600 mm. The δ-ferrite contents of the cold rolled samples and the elongated samples were compared based on the equivalent deformation. Measuring lengths were used to divide the elongated sample to equal volumes before the elongation. This division was necessary for the calculation of the equivalent deformation. Figure 20 illustrates the used volume division.

The equivalent deformation of one part can be calculated using Equations (10)–(13) [37,38]:(10)$$\bar{\varphi }=\frac{\sqrt{2}}{3}\sqrt{{\left(\varphi_{x}-\varphi_{y}\right)}^{2}+{\left(\varphi_{x}-\varphi_{z}\right)}^{2}+{\left(\varphi_{y}-\varphi_{z}\right)}^{2}}$$
(11)$$\varphi_{x}=ln\frac{a}{a_{0}}$$
(12)$$\varphi_{y}=ln\frac{b}{b_{0}}$$
(13)$$\varphi_{z}=ln\frac{c}{c_{0}}$$
where “*a*_0_,” “*b*_0_”, and “*c*_0_” are the original sizes of one part; “*a*”, “*b*”, and “*c*” are the deformed sizes of one part measured after the elongation. The equivalent deformation of the cold rolled samples can be calculated similarly by Equation (10). After the heat treatment, the δ-ferrite content of the cold rolled samples and the elongated samples were measured. Figure 21 represent the δ-ferrite contents in function of the equivalent deformation by the two different plastic deformation technologies.

It can be seen that the δ-ferrite content reduction of the elongated samples is very similar to the results of the cold rolled samples. The stronger the previous deformation extent, the higher the amount of the decomposed δ-ferrite. The single-axis tensile stress can influence the eutectoidal phase transformation process similarly to that of the stress state that forms during the cold rolling.

## 4. Summary and Conclusions

The eutectoidal phase transformation of 2507-type SDSS was examined in this complex study. The effect of the previous cold working and heat treatment was studied. Samples were differently cold worked and heat treated in three sets.

The first sample set was cold rolled at seven different rolling reductions up to 61.9%, which was followed by heat treatment at five different temperatures up to 850 °C.

The second sample set was prepared to determine the activation energy of the eutectoidal decomposition. These samples were cold rolled at seven different rolling reductions and were heat treated at 850 °C during eight different terms.

The third sample set was elongated by single-axis tensile stress and were heat treated at 850 °C. 

The δ-ferrite contents were calculated using the results of AC and DC magnetometer measurements. Because the coercivity of the studied SDSS samples was relatively high, the AC magnetometer was not able to saturate them magnetically. Therefore, the DC magnetometer was used in the following part of this work for determining the δ-ferrite content. The accurate value of the coercive field was determined by a DC coercimeter device. Light microscope and EBSD images clearly demonstrated the process of the eutectoidal decomposition. The thermoelectric power and the hardness of the samples were also measured. 

The results of all these measurements demonstrated the decomposition of the δ-ferrite into σ-phase and secondary austenite with the same tendency. It was clearly demonstrated that previous cold rolling before heat treatment significantly increases the rate of eutectoidal decomposition and decreases its starting temperature.

It was demonstrated that the thermoelectric power measurement (TEP) is sensitive to the decomposition of δ-ferrite phase. The TEP measurement is definitely a useful method for detecting the δ-ferrite decomposition process. 

The results of the magnetic measurements were compared with the results of TEP tests. It was concluded that up to 750 °C there is no significant δ-ferrite decomposition. Under 750 °C the previous cold working has no effect on the eutectoidal phase transformation. δ-ferrite decomposition was detected at 800 and 850 °C, whereby the previous plastic deformation strongly increased the rate of phase transformation. 

The activation energy of the δ-ferrite decomposition process was determined from the data of the second sample set. The obtained activation energy for the undeformed sample was 302 kJ/mol, and its value decreased due to previous cold rolling to 296 kJ/mol. These values are close to the activation energy values of the diffusion process of chromium and molybdenum in ferrite. Therefore, it can be concluded that the rate-limiting step of the whole diffusion-controlled phase transformation of δ-ferrite is the diffusion of the two mentioned alloying elements. 

The equivalent deformation rates of the cold rolled and single-axis elongated sample sets were compared. It was demonstrated that elongation had a stronger effect on the δ-ferrite decomposition process in the 0%–70% equivalent deformation range than the cold rolling. Above 70% deformation, the two ways of plastic deformation had the same effect on the δ-ferrite decomposition process.

## Figures and Tables

**Figure 1 materials-12-02205-f001:**
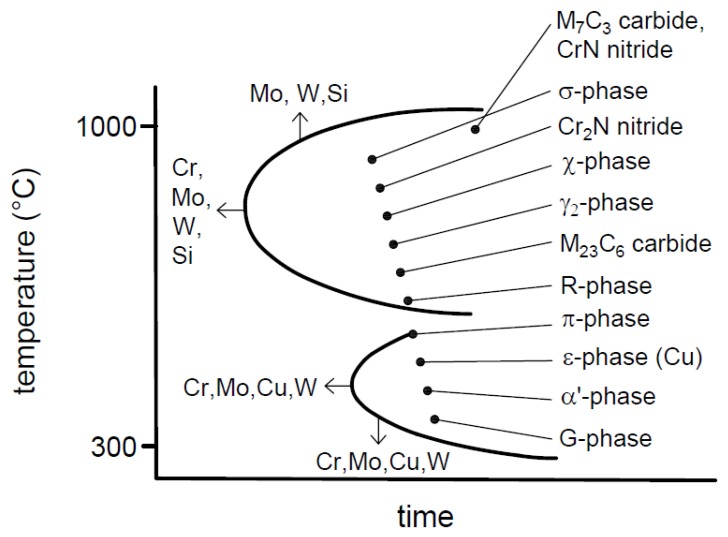
Typical precipitations of duplex stainless steel.

**Figure 2 materials-12-02205-f002:**
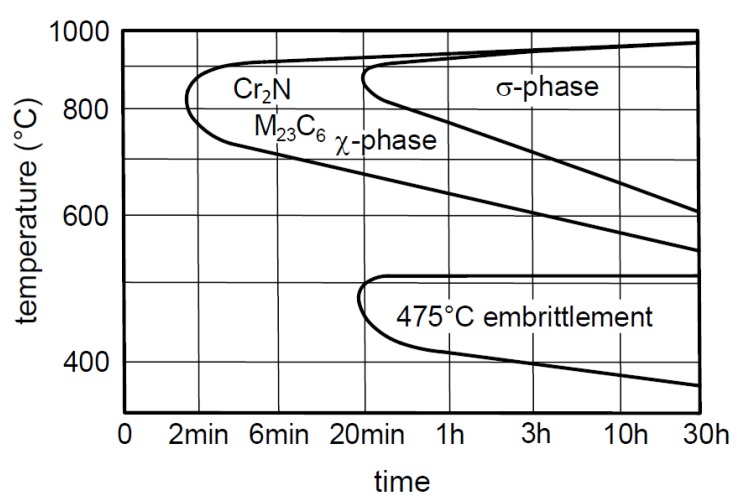
G. Herbstleb and P. Schwaab’s simplified precipitation diagram for duplex stainless steel.

**Figure 3 materials-12-02205-f003:**
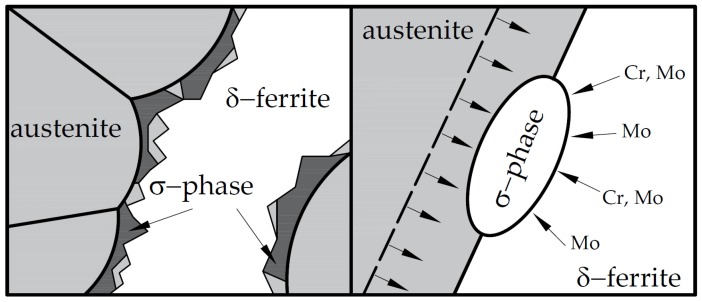
Kinetics of the eutectoidal decomposition of δ-ferrite in DSS.

**Figure 4 materials-12-02205-f004:**
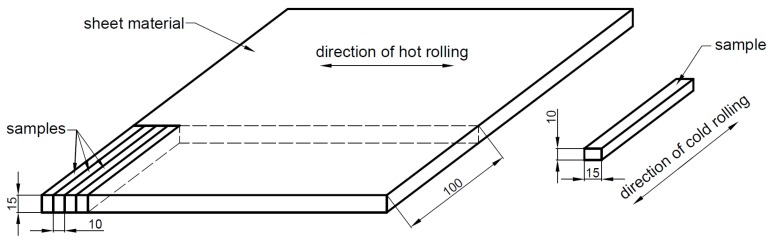
Directions of the manufacturing hot rolling and the experimental cold rolling.

**Figure 5 materials-12-02205-f005:**
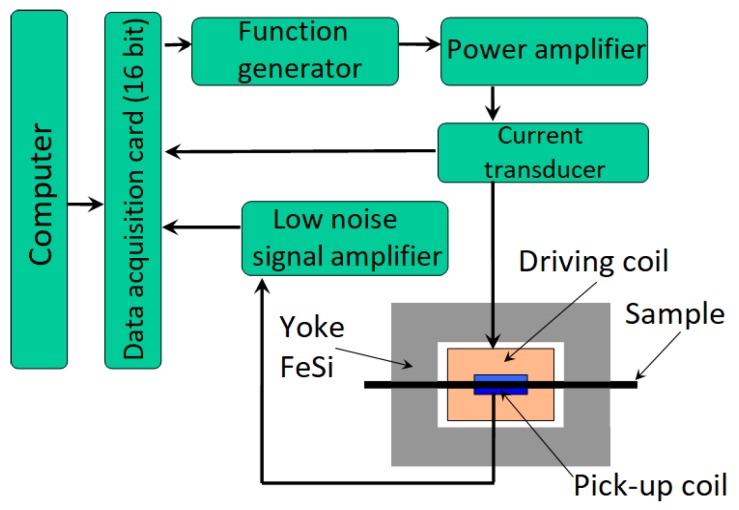
Set-up of the AC magnetometer.

**Figure 6 materials-12-02205-f006:**
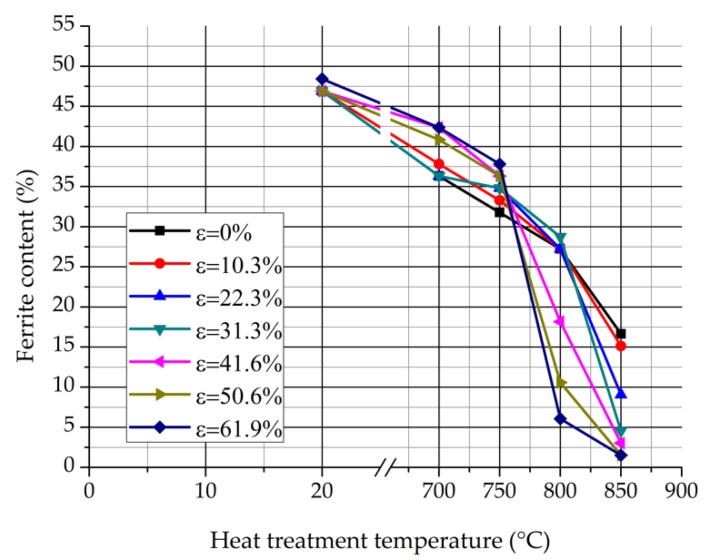
δ-ferrite contents in function of the heat treatment temperature measured by the AC magnetometer.

**Figure 7 materials-12-02205-f007:**
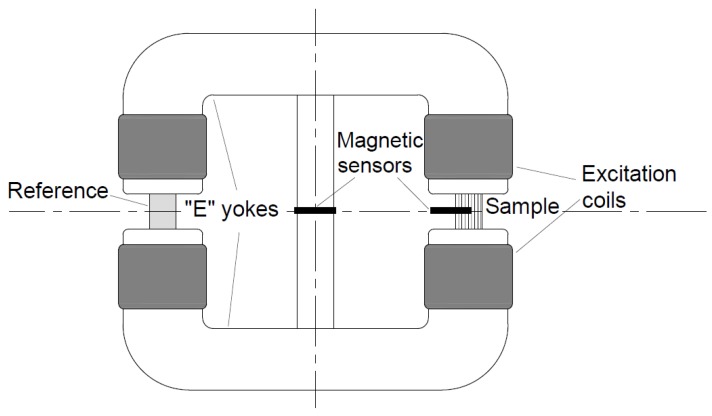
Set-up of the DC magnetometer.

**Figure 8 materials-12-02205-f008:**
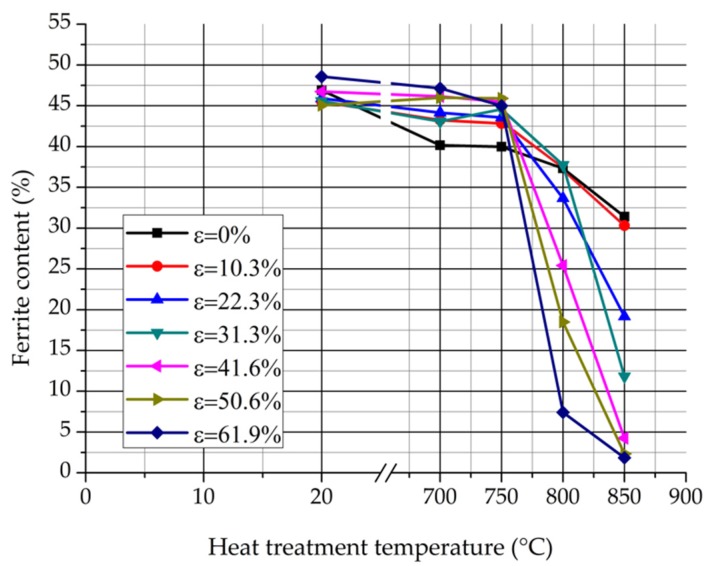
δ-ferrite contents vs. the heat treatment temperature measured by the DC magnetometer.

**Figure 9 materials-12-02205-f009:**
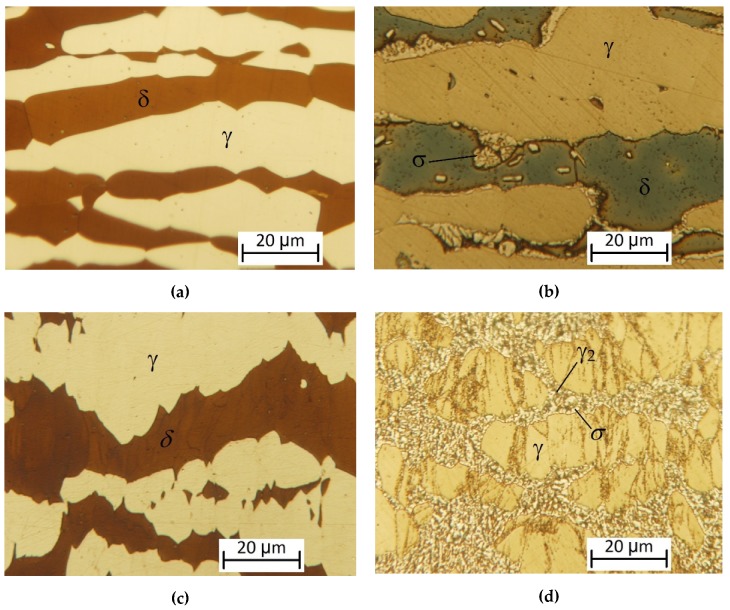
Microscope images of 2507-type SDSS in the magnification of 1000×: (**a**) Base microstructure of the 2507-type SDSS; (**b**) heat-treated sample at 850 °C without deformation; (**c**) ε = 61.9% deformed sample without heat treatment; (**d**) ε = 61.9% deformed sample and heat treated at 850 °C.

**Figure 10 materials-12-02205-f010:**
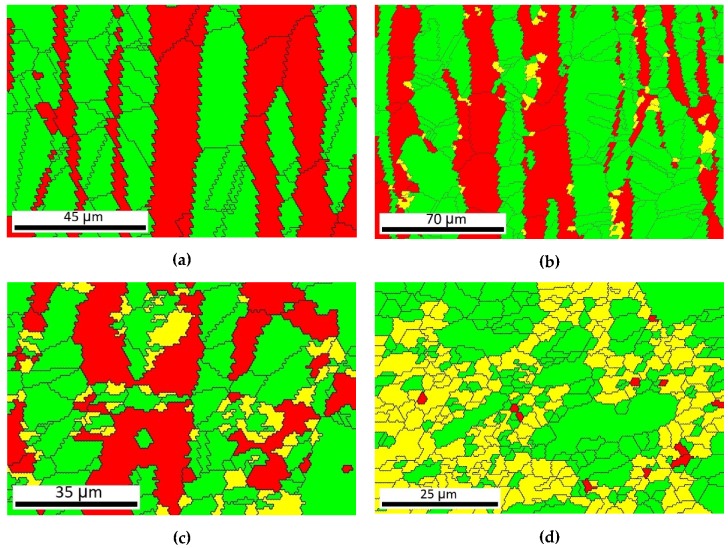
Phase maps made by EBSD: (**a**) The original phase ration of the 2507-type SDSS; (**b**) heat-treated sample at 850 °C without deformation; (**c**) ε = 22.3% deformed sample and heat treated at 850 °C; (**d**) ε = 61.9% deformed sample and heat treated at 850 °C. (Color marking: red area—δ-ferrite; green area—austenite; yellow area—σ-phase).

**Figure 11 materials-12-02205-f011:**
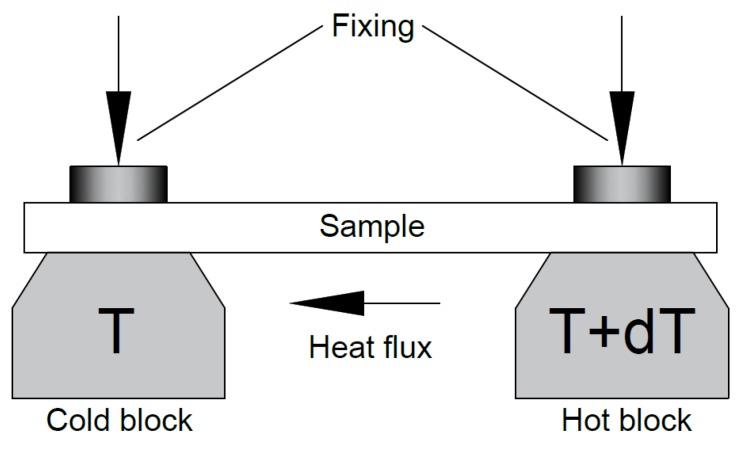
Sematic illustration of the TEP instrument.

**Figure 12 materials-12-02205-f012:**
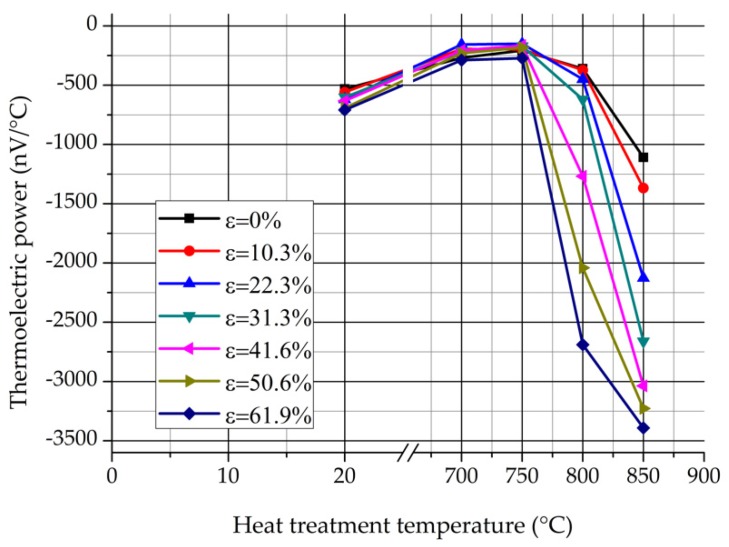
Values of the TEP in function of the heat treatment temperature.

**Figure 13 materials-12-02205-f013:**
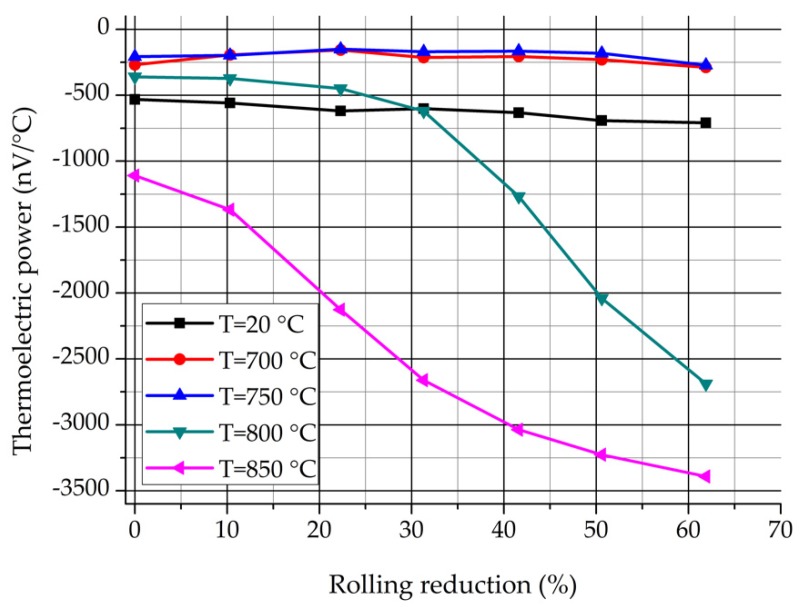
Values of the TEP in function of the rolling reduction.

**Figure 14 materials-12-02205-f014:**
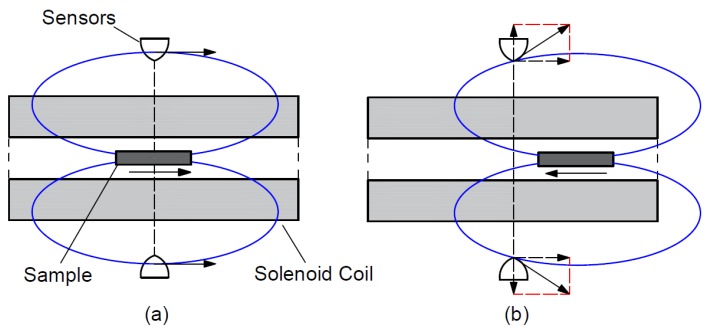
Set-up of the Foerster-type DC coercimeter: (**a**) First step of the measuring; (**b**) second step of the measuring.

**Figure 15 materials-12-02205-f015:**
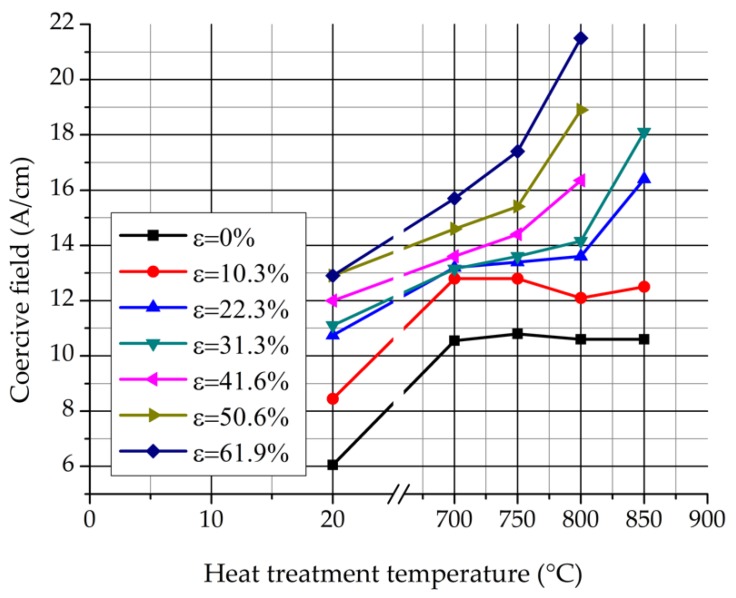
Values of the coercive field in function of the heat treatment temperature.

**Figure 16 materials-12-02205-f016:**
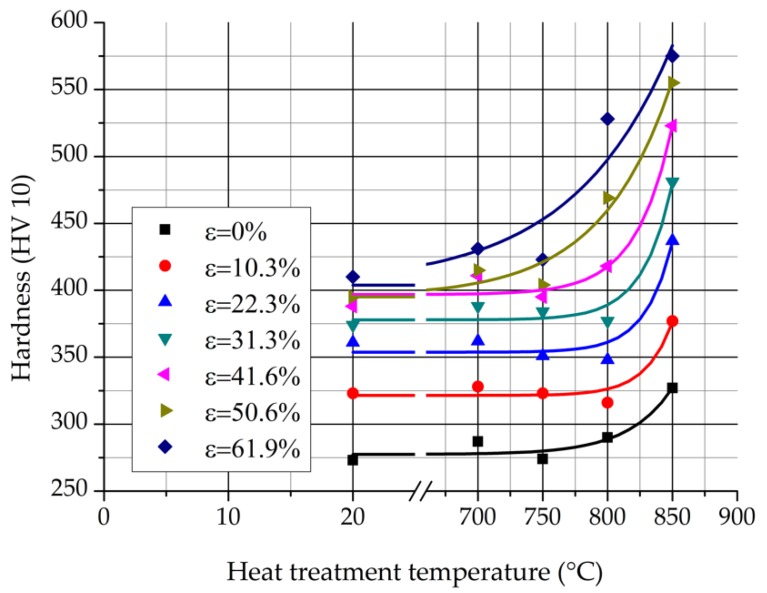
Hardness of the samples in function of the heat treatment temperature.

**Figure 17 materials-12-02205-f017:**
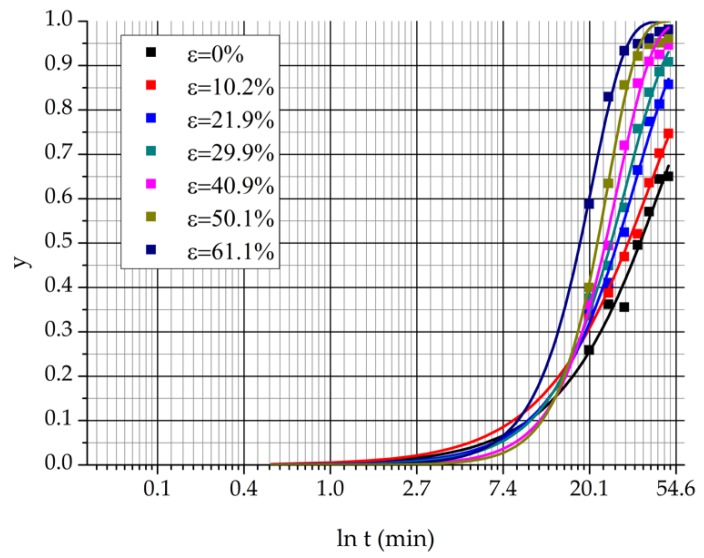
Avrami curves.

**Figure 18 materials-12-02205-f018:**
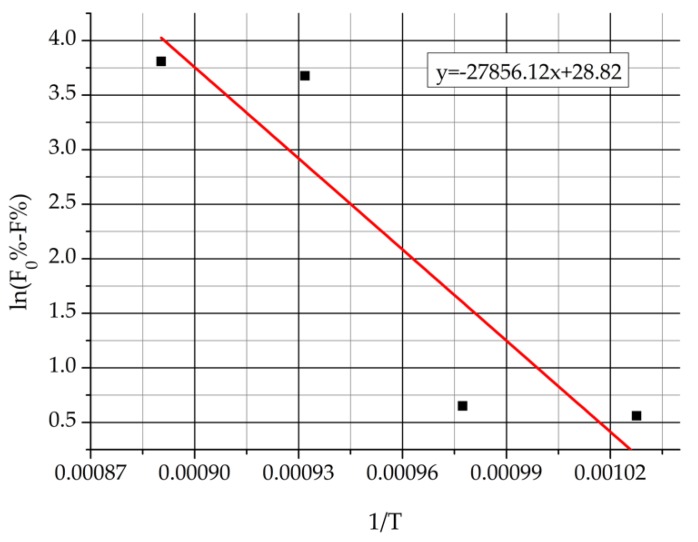
Arrhenius plot of the ε = 61.9% deformed original sample series.

**Figure 19 materials-12-02205-f019:**
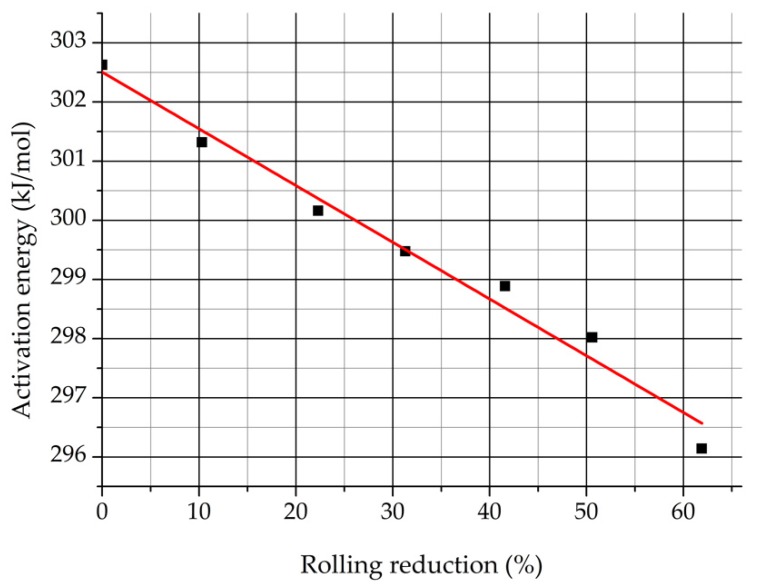
Activation energy in function of the rolling reduction.

**Figure 20 materials-12-02205-f020:**
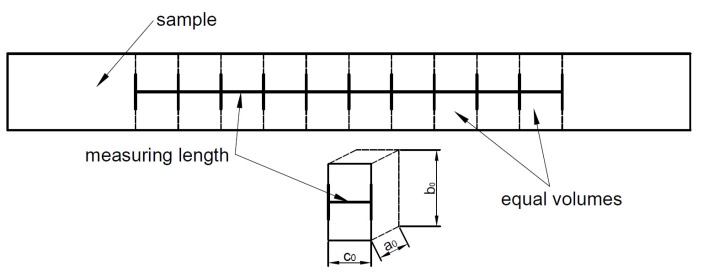
Volume division of the sample before the elongation.

**Figure 21 materials-12-02205-f021:**
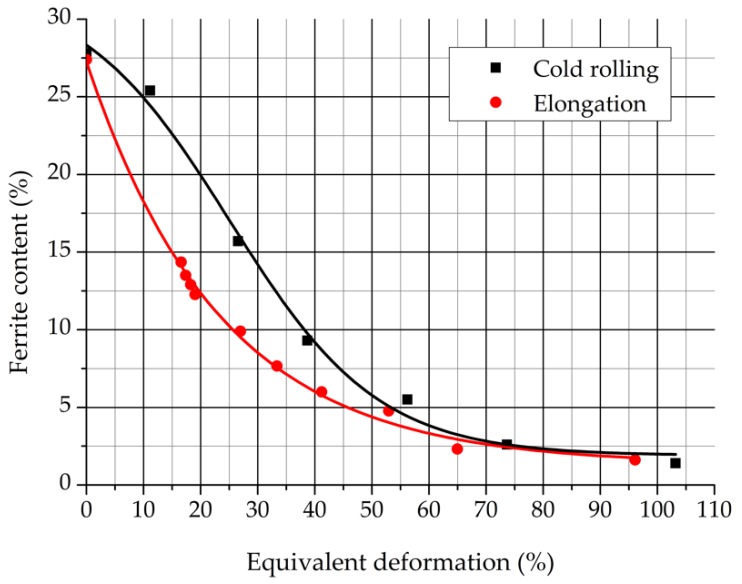
The equivalent deformation in function of the δ-ferrite content.

**Table 1 materials-12-02205-t001:** Chemical composition of duplex stainless steel (DSS) (%) [3].

C Max.	Cr	Ni	Mo	W	Cu	N
0.03	22–25	4–7	0–4	0–2	0–1.5	0.1–0.35

**Table 2 materials-12-02205-t002:** Chemical composition of the 2507-type SDSS (%).

C	Mn	P	S	Si	Cu	Ni	Cr	Mo	Nb	Ti	N
0.021	0.822	0.023	0.0004	0.313	0.178	6.592	24.792	3.705	0.008	0.005	0.264

**Table 3 materials-12-02205-t003:** Mechanical properties of the 2507-type SDSS.

Yield Stress Rp_0.2_ (MPa)	Tensile Stress Rm (MPa)	Elongation at Fracture A (%)
634	829	26

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
