# Peer review of "Complex Study of Eutectoidal Phase Transformation of 2507-Type Super-Duplex Stainless Steel"

_materials, 2019, doi:10.3390/ma12132205_

Round 1

Reviewer 1 Report

The paper is well prepared, the topic related to duplex steel is always interesting to the readers. Due to this it absolutely deserves publishing.The paper contains minor spelling or tense mistakes regarding to English. Some minor changes like the separation of the equation (7) would improve the readability. The description of eq.(7) could be added in regards to a citation, because we shouldn't describe it as the well known equation. The conclusion part could be improved in the sense of adding data that would support them. Overall the reviewer suggests accept after minor revision. 

Reviewer 2 Report

372 – 382 – Why you didn’t use the specimen shape according to DIN ISO 6892-1? The used shape is bad for a trustful determination of elongation (deformation). Can you guarantee that there is no deformation at the fixation? Was the deformation uniform over the full length of the sample?

The whole paper is well written. But what is propose? I mean you did 8 different experiments to show the same. There is no deeper look into the process of the eutectoidal phase transformation. The paper is a summery of different analyse methods not more. Please reconsider what you want to show. 19 pages for nearly the same result.

According to the Si unit system the unit should be separated by a space bar form the value.

Reviewer 3 Report

Summary:

The isothermal ferrite decomposition of 2507 DSS is studied.

In first part, the effect of cold rolling induced deformation ~0-61% reduction on the ferrite decomposition during 30 min isothermal heat treatment at 700,750,800 and 850 C is addressed by measuring residual d-ferrite using magnetic methods. The measurements are complemented with thermoelectric power, coercivity and hardness measurements.

In second part, the transformation kinetics is studied by isothermal heat treatment at 850 oC for 20-50minutes and activation energy as a result of cold rolling reduction is deduced.

In third part the deformation induced during tensile test is related to remaining d-ferrite after heat treatment at 850C for 30 min.

Recommendation:

The  article contains significant amount of interesting experimental data, however the decription of used method, interpretation and discussion of results and numerical processing is rather poor. Therefore the reviewer recommends MAJOR REVISION of the article based on the following comments:

Major comments

A.      The article is in fact separated in three parts, please clearly mark this separation in text and describe the aim of each part (see above summary) (the description in abstract is insufficient as no aims are presented)

B.      The article uses three methods to estimate ferrite (maximum polarization, saturation polarization and ‘ferriscope’ which is uses unspecified magnetic method).

B1 - The difference between the maximum and saturation polarization method should be discussed, i.e. why to present maximum polarization data if more adequate saturation data are also presented.

B2 - The Ferriscope should be described (at least make, model and probe)

C.      The  study of transformation kinetics and activation energy is very hard to follow             

C1 – Avrami equation can be used to get the maximum transformation rate k. Replacing rate k by the proportional fraction of transformed ferrite adds another constant P, i.e. k=P(y-y0) to eq 6. Still it is possible to get Ea as a slope from Fig 18. However as single temperature was used, data from part 1 must be used to construct fig. 18 , this must be mentioned. Was the ferrite content remeasured using ferriscope  ?

C2 – All equations follow eq. 2 and 3 using simple algebra, this must be rigorously mathematically presented in the paper including the assumptions (both fulfilled and not fillfilled).

D.      The results should be discussed against available literature on the topic

E.       The “simple proportion” should be formalize mathematically, if possible, samples measured by saturation polarization should be compared to ‘ferriscope’  and maximum polarization.

Minor comments

L20-23 – add aim and structure the abstract in the three parts addressed in paper

L116 – five-five

Figure 6. Heat treatment, consider splitting x axis in two parts and removing the portion from 20 to 700C

L192 – saturation gives more accurate results, why present maximum polarization data then ?

L196 – optical -> light

L197 – metal -> metallographic

L200 and elsewhere -  used equipment should be presented in standard way (make, model, parameters, probes etc)

Fig 9-10 – please describe phase on all images clearly

L302  - specify load

Fig. 17 – how were the curves fitted ?

Round 2

Reviewer 2 Report

The changes provide a better understanding about the choose of experiments. Only one last thing I must complain. There must be a space between numbers and percentages, too. Good work.

Reviewer 3 Report

The authors adressed all reviewers comments and the article is suitable for publication in reviewers opinion